# Design of a Highly Efficient Wideband Multi-Frequency Ambient RF Energy Harvester

**DOI:** 10.3390/s22020424

**Published:** 2022-01-06

**Authors:** Sunanda Roy, Jun-Jiat Tiang, Mardeni Bin Roslee, Md. Tanvir Ahmed, Abbas Z. Kouzani, M. A. Parvez Mahmud

**Affiliations:** 1Faculty of Engineering (FOE), Multimedia University, Persiaran Multimedia, Cyberjaya 63000, Malaysia; jjtiang@mmu.edu.my (J.-J.T.); mardeni.roslee@mmu.edu.my (M.B.R.); tanvir1533@gmail.com (M.T.A.); 2School of Engineering, Deakin University, Waurn Ponds, Geelong, VIC 3216, Australia; abbas.kouzani@deakin.edu.au (A.Z.K.); m.a.mahmud@deakin.edu.au (M.A.P.M.)

**Keywords:** ambient, broadband rectifier, log periodic antenna, quad band, RF energy harvesting, RF spectral survey

## Abstract

For low input radio frequency (RF) power from −35 to 5 dBm, a novel quad-band RF energy harvester (RFEH) with an improved impedance matching network (IMN) is proposed to overcome the poor conversion efficiency and limited RF power range of the ambient environment. In this research, an RF spectral survey was performed in the semi-urban region of Malaysia, and using these results, a multi-frequency highly sensitive RF energy harvester was designed to harvest energy from available frequency bands within the 0.8 GHz to 2.6 GHz frequency range. Firstly, a new IMN is implemented to improve the rectifying circuit’s efficiency in ambient conditions. Secondly, a self-complementary log-periodic higher bandwidth antenna is proposed. Finally, the design and manufacture of the proposed RF harvester’s prototype are carried out and tested to realize its output in the desired frequency bands. For an accumulative −15 dBm input RF power that is uniformly universal across the four radio frequency bands, the harvester’s calculated dc rectification efficiency is about 35 percent and reaches 52 percent at −20 dBm. Measurement in an ambient RF setting shows that the proposed harvester is able to harvest dc energy at −20 dBm up to 0.678 V.

## 1. Introduction

Due to its simplicity in construction, self-powered low-power electronics devices have attracted remarkable attention for numerous wireless applications such as healthcare sensors, smart cities, Internet-of-Things (IoT), actuators and wireless sensor networks (WSNs) in the last couple of years. Due to the importance of batteryless devices and maintenance cost-related issues in the conventional battery-operated system, the market for such energy scavenging technologies is growing continually. Despite significant developments, the battery’s lifespan is still limited, and their replacement is always complicated, requiring regular maintenance. In addition, in the power conversion process, a large amount of energy is wasted for charging various electronic systems such as mobile phones, tablets, smartwatches, e-readers, pin pads and other sensing components. Energy harvesting techniques from different energy sources (i.e., radio frequencies, solar, and thermal) available in an ambient environment can solve this issue [1]. RF energy harvesting is the most favourable technique of converting microwave energy to dc electrical energy from the ambient environment, the concept of which was developed in the early-1990 era [2]. The key components of the RF energy scavenging system are depicted in Figure 1.

Most other energy sources are predominantly dependent on the environment, except for RF energy, which makes this technique more fruitful in various critical applications, including smart monitoring systems, healthcare, household appliances, protection, etc. [2,3] Wireless energy harvesting (WEH) and wireless power transfer (WPT) have gained significant attention as well as perceived immense development over the last few years [4,5,6]. The rectifying circuit is a key device for converting RF to DC energy with high-level conversion efficiency for radiative and inductive wireless power transmissions, respectively [7]. One of the key devices for both WPT and WEH applications is the broadband rectenna, and enormous progress has been achieved [4,5,6,8,9,10,11,12,13]. The broadband multi-frequency rectenna consists of a multiband rectifier circuit and a multiband antenna [4,5,6,12,13] which are very crucial to harvesting energy from numerous RF sources and different frequency channels simultaneously. They thus perform better in terms of overall RF-to-DC rectification efficiency as well as overall output DC power than single band rectenna [6,8,9,10]. Additionally, the broadband multi-frequency harvester impedance matching network is as daunting as an antenna. The impedance matching network (IMN) design is comparatively complex, increasing the cost and failure and familiarizing itself with manufacturing errors. Another crucial problem in RF energy scavenging is the very low input RF power density level of indoor and outdoor ambient environments, either in rural or urban areas. This obstructs the RF power received by the antenna and conducts low RF-to-DC conversion efficiency and low output dc voltage of the rectifier. Some RF harvesters have already been designed where the RF-to-DC rectification efficiency is greater than 80%. However, most of these rectennas were the single band and were able to harvest energy with higher input RF power densities such as 10 dBm [14], 0 dBm [15], 13 dBm [11], and 6 dBm [15], which is not realistic for ambient energy harvesting.

However, a rectenna’s key importance is to design a multi-frequency wideband antenna because it can harvest RF energy from over all frequency bands concurrently available in an ambient environment. Additionally, a multi-frequency broadband rectifier is also necessary to convert DC energy for all captured signals by the multiband antenna. Moreover, the non-linear rectifier circuit’s input impedance varies with different frequencies, input RF power density level, and load impedance, respectively. Some dual-band harvesters, multiband harvesters, and broadband harvesters have been designed [4,5,6,12,16,17,18,19,20], but there are several limitations. In this paper, a wideband multi-frequency RF harvester (i.e., wideband log-periodic antenna and broadband rectifier including a novel RF filtering concept) is proposed to harvest energy from all available RF energy sources that exist in the ambient environment. To illustrate the feasibility of implementing a multiband ambient RF energy harvesting technique, we initially depicted an RF power spectral survey’s detailed results in this research, specifying suitable urban locations and available frequency bands with associated RF input power levels for scavenging. Based on these RF survey results, a multi-frequency highly sensitive RF harvester is then designed and fabricated, and its performances under energy scavenging operation are considered employing in situ field strength measurements.

The rest of this article is structured as follows. Section 2 presents a new strategy of RF spectral survey in the urban ambient environment in Malaysia. A self-complementary dual arms log-periodic wideband multi-frequency antenna is designed, implemented and analyzed in Section 3. A broadband multi-frequency rectifier is designed and analyzed in Section 4. The investigational results of the RF harvester including indoor and outdoor environmental measurements are observed in Section 5. Section 6 discusses the observation of the proposed RF harvester with bq-25504 power evaluation module integration. Lastly, the concluding remarks are depicted in Section 5.

## 2. MMU RF Spectral Survey in Malaysia

To determine available frequency bands with associated input RF power density levels in a typical urban ambient environment, an RF spectral survey within the frequency range of 0.8 to 3 GHz is conducted in the MMU campus at Cyberjaya, Malaysia. In a semi-urban ambient environment, the RF spectral survey may not provide enough information because the RF power density of most of the base-stations changes with local geographical variations and different propagation characteristics, such as multipath fading, diffraction and attenuation, and so on. Each building on the MMU area is used as an RF survey point to provide detailed information for representing Malaysia as a function of geographical distribution and population density, having an amalgamation of rural and urban characteristics. A new RF survey technique is introduced to perform an RF spectral survey for each survey point shown in Figure 2a. Several RF spectral surveys depend on the number of survey points (i.e., number of buildings existing in the MMU area). In Figure 2b, the red exploration signs are indicated by the number of RF survey points at MMU. It is clearly stated that various buildings such as the Faculty of Engineering (FOE), Digital Library (DL), Student Service Centre (SSD), Institute for Postgraduate Studies (IPS), Faculty of Computing and Informatics (FCI), Faculty of Management (FOM), Central Food Corner (CFC) and MMU Mosque are considered as RF spectral survey points of the entire MMU campus. By following each of the survey points, the RF spectral survey is started from a very close location to the points, that mean 0 m, 25 m, 50 m and 75 m, respectively.

The RF survey is performed in a clockwise direction (dotted sing curve of Figure 2a) and maintained at least 90 degrees apart from each other. The number of RF surveys in each survey point is depended on the area occupied by the building or survey point. The overall university is covered by applying the same process of RF spectral survey. Additionally, few more RF surveys are taken inside the playground and different accommodation blocks (i.e., HB1, HB2, HB3 and HB4) of MMU. Cellular phones’ application differs throughout the daytime. Therefore, ambient RF power in their using frequency bands is expected to be time-dependent; available RF energy density level is more significant in the daytime than at night. Consequently, to make real comparisons between survey areas, measurements are taken between 9:00 a.m. and 4:00 p.m. on 7 days in a week. A TTi PSA6005 Handheld Spectrum Analyser (Manufacturer Thurlby Thandar Instruments Ltd, City Cambridgeshire, United Kingdom) with a calibrated omnidirectional antenna has been used to measure electric field strength from the 0.5 GHz to 3 GHz frequency range. Here, the calibrated antenna is rotated to different axes and waited while the RF spectrum analyzer is set to the max power density level, ensuring that the highest reading is recorded. More than 1 or 2 min were assigned for each reading to allow for more than two sweeps across the target frequency band. Moreover, to maintain an equal signal-to-noise (S/N) ratio, attenuation is familiarized (with at least 5 dB) to get rid of compression when maximum input RF power density levels are detected. For each measurement, the analyzer settings are carefully chosen with the 100 kHz resolution bandwidth (BW) and peak resolution of 1001 points to obtain a snapshot of the power density level in an urban area from unpredictable RF sources. To ensure a perfect calibrated system, electric field strength and RF power level measurements are taken by connecting with an analyzer and frequency banded antenna. Figure 3 illustrates the RF input power density level measured within the area of MMU, where the available frequency bands for GSM 900, GSM 1800, 3G, Wi-Fi and LTE can be identified. A highly sensitive multi-frequency broadband RF harvester can harvest energy across all frequency bands, and thus it is important to calculate the overall band power density level. The overall RF survey overview (i.e., available frequency bands with associated RF input power level) within the MMU campus is depicted in Table 1. It demonstrates that comprising of UpLink (UL) and Down Link (DL) modes, the GSM 900 full frequency band is from 0.876 to 0.959 GHz. Similarly, GSM1800 is including UL and DL modes ranging from 1.71 to 1.88 GHz. The UMTS band is between 1.920 and 2.17 GHz (comprising UL and DL modes), and the Wi-Fi band is between 2.40 and 2.45 GHz at the end. The information is used as an ambient harvester design base point since the RF power level at each frequency band will define a multiband rectenna’s input impedance. Using the detailed information from the MMU RF spectral survey, Table 1 depicts the average and median of the available ambient frequency band with associated input RF power level measurements for Malaysia’s four largest ambient RF energy sources.

## 3. Design Facets of the Suitable Antennas Geometry for RFEH System

On an FR4 substrate material with a relative permittivity ($\varepsilon_{r}$) of 5.4, loss tangent (tan δ) of 0.02, and a height (h) of 1.6 mm, the suggested self-complementary log-periodic frequency-independent toothed planar multiband antenna is made. The overall dimension of the substrate is 160 mm × 1 60 mm, which equals 0.29λ_0_ × 0.29λ_0_ at 550 MHz as shown in Figure 4. It is an extension of the bow-tie antenna with all teeth circularly spread on both sides. The antenna consists of 12 radiators (called teethes of the antenna) and the apex angle of each tooth is 45° where the boom angle is 20.4°. The tooth’s mth number is demonstrated by an outer radius $R_{m}$ and inner radius $r_{m}$ defined as [22].
(1)$$\frac{R_{m}{}_{+1}}{R_{m}}=\frac{r_{m+1}}{r_{m}}=k=1.45$$
where the value of *R*_1_ and *r*_1_ is 70 mm and 48 mm, respectively. The antenna is coupled to a 20.04 mm width aperture line at the centre. The arm has multiple teeth with distinct measurements of radius and slots (with a common centre feeding position), as depicted in Figure 4a. The multiple teeth have spread both sides of the antenna in such a way that the right-side teeth are placed in between a set of left-side teeth, as presented in Figure 4b. The spacing between the two teeth is decreased with a decrease in the size of the teeth. This radiating antenna element is arranged in such a way that the upper and lower section complementary along the horizontal axis. Impedance matching is one of the key criteria in frequency-independent log-periodic antennas. For smoothening the external beam former and making the antenna more directional, the complementary aperture has perfect integration with all arm teeth alongside the vertical axis to maintain 50 Ω input impedance.

The matching network changes the nominal impedance of the aperture line to 50 Ω while working as a balun, as well as modifies the external beam former to a single 1800 hybrid for dual-CP usage. The geometry and layout of the proposed frequency-independent log-periodic antenna are highlighted in Figure 4. Table 2 displays the various parameters of the proposed, such as the width of the slot or tooth W and the size of arc radius R, aperture angle, boom angle, gap, length of aperture, etc. The evolution of the antenna’s basic structure with its various radiating elements is shown in Figure 5. In the main radiating element, twelve radiators or teeth are produced to increase the impedance bandwidth to cover all frequency bands. The log-periodic (LP) antenna design principles are used as a basic process of the proposed antenna design and then further optimized with commercially available high-frequency EM simulation software package “Computer Simulation Technology (CST)” v.19.

To improve the overall antenna performance (i.e., gain, BW, radiation efficiency), the core radiating element is arranged in a font-to-font fashion that looks self-complementary. Finally, with a gap of 0.25 mm, an annular circle is built around the self-complementary radiating portion. This annular configuration is connected to the antenna edge and serves as a lower frequency radiator [23]. The full annular shape is connected to the antenna’s other rings and functions at the lower resonance frequencies as a radiator. One arm of the antenna is attached to the inner conductor of the 50 coaxial cables, while the outside conductor is linked to another arm that is called ground metal. Figure 6 shows the fabricated prototype images (i.e., front and back sides) of the suggested LP antenna.

## 4. Broadband Performance of LP Antenna

S-parameter (S11) expresses the connection among input and output ports in the transmission of electric system. It represents the excellence of the impedance matching between the input source and the transmission line’s output load. For any proposed antenna, the value of the reflection coefficient (S11) is considered when it is less than −10 dB [24]. The comparison between simulated and measured scattering parameter S11 [dB] versus frequency of three distinct design stages are depicted in Figure 7, where overall performance is in good agreement. It is examined from Figure 7 of the S-parameter (S11) that the propagating EM wave traverses the −10 dB line thrice from 0.5 to 3.5 GHz range.

The black dash line indicates the measured return loss value, which is higher than the simulated value (indicated by the blue line) of the proposed antenna. The first resonance is found at approximately 0.5 GHz and the value of RL is approximately −15 dB. The corresponding bandwidth −10 dB of the centre frequency is 0.5 GHz. To get wider bandwidth at larger frequencies (above 1.5 GHz), the rest of the rings are modified to frequency-independent equal radius log-periodic resonator teeth or rings, as depicted in Figure 5.

The antenna covers six resonant frequencies at six different bands, and these resonances occurred in the range of 0.5 to 3.5 GHz. To determine the proposed antenna’s performance parameters, a discrete port is applied between two symmetrical arms of the log-periodic rings that produce the dual circular polarized radiation field. Additionally, the overall performance is enhanced due to the circular rings’ modification with an outside annular ring. It is proven that the dips observed in the scattering plot indicate the excellent impedance matching of the proposed log-periodic antenna. The return loss values for all resonant frequencies are quite similar except few discrepancies. The central discrete port–feeding 0.5 to 3.5 GHz complimentary ring log periodic antenna is simulated using computer simulation technology (CST) software and a comparison between simulated and measured maximum realized gain versus frequency is presented in Figure 8.

The measured co-polarized maximum realized gain is approximately 6 dBi across all RF operational frequency bands from 0.5 to 3.5 GHz. The somewhat maximum value of cross-polarized realized gain is due to the large radius (R) and a wide-angle on the antenna turnstile ring. The large radius and large angle of the complementary ring are needed to allow for the coaxial feeder line to fit except for shorting the high-frequency radiating elements. The measured and simulated cross-polarized realized gain has a slight variation in which measured gain has a higher value than that of the simulated gain at GSM 900, GSM 1800 and LTE with an exception for the case of 2.4 GHz. From Figure 9, it is noticeable that the surface current on the novel rings or teeth excites radiating elements, mainly at the resonant frequency bands of radiating elements. Moreover, it can be seen that the majority of the current particles exist between inner rings around the center and the outermost rings of the proposed geometry at 1.8 GHz, 2.38 GHz and 2.9 GHz frequency bands (except for 2.9 GHz). At the 0.78 GHz frequency band the surface current density is mostly concentrated at the peripheral region of the second and third radiating rings of the proposed antenna. It is also observed that both the symmetrical rings of the proposed structure interact to produce the multiband operation. The radiation pattern or directivity is the main concern for antenna performance. It characterizes the direction of an antenna at which the signal is radiated in a particular direction. The ratio of two radiation intensities is calculated, while one is defined in a specific direction and the other is the overall direction. The simulated and measured radiation patterns (both 2D and 3D) of the proposed LP antenna at different resonance frequencies across the bands are illustrated in Figure 10. It can be seen that the cross-pol exists within the parameter of the co-pol which indicates good antenna characteristics. From Figure 11, it is observed that the ring-turnstile part of the aperture connection dominating at 0.9 GHz; 1.2 GHz and 1.4 GHz are the frequencies in the transition region where both ring-turnstile and the LP’s outermost teeth contribute to the radiation and the LP dominates at 1.8 GHz, 2.15 GHz, 2.4 GHz and 2.9 GHz. Since the WoW (the variance in the far-field intensity at a given azimuth plane while the elevation angle remains fixed) is very small and the efficiency of the higher order mode is also poor, the stability of the directivity radiation patterns over the entire frequency band is excellent over active regions of all the aperture components.

## 5. State-of-the-Art: Multiband Rectifier Architecture for RF Harvester Technology

The conversion of RF energy (exploited from available RF sources from the ambient environment) into DC energy is the goal of a rectifying antenna or RF harvester. A standard RF harvester requires a receiving antenna and a circuit rectifier (i.e., RF filter, a rectifier section, a low-pass filter section, and a load branch). A lot of researchers have been focusing on rectenna designed for different frequency bands such as single band, dual bands, triple bands and so on [25,26,27,28]. Such RF harvesters’ functionality increases significantly if the operating frequency remains the same as the available ambient frequency. A single band, dual-band or even triple bands RF harvester is not suitable for ambient applications. Certainly, the preponderant frequency bands are significantly changed from one place to another at an ambient level. Some multi-frequency RF harvesters are reported in [6,29,30,31,32,33,34]. These RF harvester advantages from the accumulation of several RF frequency bands ensure the scavenging of a higher amount of dc energy [35].

### 5.1. Multiband Rectifier Topology

Several rectenna topologies are investigated to realize the energy harvesting technique from multi-frequency bands. The design of an RF bandpass filter is the key element that contains the main difference. The RF filter functionality determines the matching between the antenna and the rectifier input impedance. The input impedance of the rectifier differs with the variation of the number of frequencies and incident power. Moreover, the antenna impedance is a function of frequency. It varies with the variation of frequency bands. Therefore, it is easier to impedance match at a single frequency rather than multi-frequency. Based on these criteria, there are different kinds of losses, while one (impedance mismatch) occurs due to the RF filter circuit’s complexity. These losses due to impedance mismatch over a wide bandwidth are explained in [30,32]. The architectural details of the multiband RF harvester are shown in Figure 11. The variation of a few hundred MHz of an RF band causes impedance disparity, and hence a decline of the RF-to-DC rectification efficiency. The RF-to-DC rectification efficiency is about 8% in the RF band of 1.5 GHz for a harvester topology with an incident input power density of −20 dBm [32]. To obtain perfect impedance matching between the receiving antenna and rectifier circuit, the losses caused by the filter elements are crucially important. This type of topology is not selected for harvesting energy from the entire RF frequency bands. Finally, energy harvesting from multi-frequency bands can be possible by using stacking some harvesters. In this situation, the RF filter circuit is designed using a specific center frequency and the harvesters generated dc voltages connected to the identical load [33,34]. In most research papers, the combination of dc output voltage is not considered for quality assessment for this topology. About 45% dc rectification efficiency is achieved at 0.9 GHz and 1.8 GHz frequency band for 15 dBm RF input power, and 25% at 2.45 GHz for the same incident RF power as demonstrated in [34].

Instead of these appealing results, the total RF-to-DC efficiency is not considered if the dual or triple RF tones are available. The major limitation of this architecture is that it cannot harvest energy outside the operating frequency bands. Due to several antennas employed in this type of harvester, it is not suitable for compact applications. Figure 12 illustrates the proposed architecture in this article. The RF branch in the proposed topology consists of the RF filter circuit, the rectifier section, and the low-pass filter section. It has no limit to increasing the number of RF sections. A single receiver antenna (i.e., multi-frequency broadband antenna) is connected to the proposed topology input section which turns into a more compact and reliable structure. To increase the dc rectification efficiency, the circuit’s impedance matching network must compose various components such as parallel rectifiers section and antenna part. The RF filter sections are used to match per parallel rectifier section at the different dedicated frequency bands. The impedance matching network is designed in such a way that it maintains broad bandwidth (BW). To cover entire frequency bands (i.e., GSM 900, GSM 1800, 3G and Wi-Fi) with associated input RF power level, a specific BW is chosen for each RF section. Finally, the dc output from each RF branch is connected to increase the overall harvested power. The low pass filter section in each branch must block RF signals and enable only the dc portion to be passed on to the output parts.

### 5.2. Elementary Discussion about Rectifier Circuit

There are different types of rectifier circuits (i.e., series, parallel and Greinacher) depending on different incident frequency bands with associated RF power density [6,31,32,33,34,35,36]. However, Greinacher rectifier topology generates a higher dc output voltage level than other topologies for given input frequencies with the associated RF power level. Moreover, the overall RF-to-DC rectification efficiency is significantly affected by the combination of dc output voltage from several RF branches [32]. The total harvested power with series connection is greater than the parallel connection [37]. Additionally, voltage equalization in series connection is better than the current equalization in parallel connection [38].

These characteristics are caused by the fact that a parallel joining creates more interference between the RF branches with dc output equated to a series joining [37,38]. In this context, a promising candidate for its differential dc output is the Greinacher rectifier circuit. As shown in [6], the dc output voltages of many stacked Greinacher rectifiers can be summarised without causing dc interference between each RF branch. The Greinacher rectifier uses diodes that are two times larger than a voltage doubler structure and four times larger than a sequence rectifier, despite having interesting features. Therefore, compared to series or parallel rectifiers, higher incident RF power is needed to turn on the Greinacher rectifier circuit’s many diodes. Due to the RF power density levels’ variation, it is necessary to limit the number of diodes for a smooth operation in an ambient environment.

### 5.3. Proposed Rectifier Topology

The selected rectifier schematic must have a restricted number of diodes. Observing the present rectifiers, only series or parallel doubler rectifier topology is suitable for ambient operation. This rectifier advantages from a differential dc output as required by defeating the voltage double link to the ground. The modified rectifier is demonstrated in Figure 13. In Figure 13, the rectifier consists of two diodes mounted in parallel with each other while only one diode starts as a threshold voltage. During the positive half cycle of the input signal ($RF_{@input}$), the diode $D_{1}$ turns off and $D_{2}$ turns on. Then, the capacitor $C_{2}$ is charged over $D_{2}$ and stored in $C_{1}$. Similarly, for the negative half cycle of the input signal ($RF_{@input}$) $D_{1}$ switch on and $D_{2}$ switch off. Hence, the capacitor $C_{1}$ is charged over $D_{1}$. Finally, the resultant differential output voltage ($V_{diff}$) equals two times of input signal ($RF_{@input}$). Both diode threshold voltages (i.e., $V_{T1}$ and $V_{T2}$) are taken into account for determined the differential output voltage ($V_{diff}$) as follows
(2)$$V_{diff}=V_{out(+)}-V_{out(-)}=2RF_{@input}-V_{T1}-V_{T2}$$

### 5.4. RF Filter Design Including Matching Element

According to the maximum power transfer theorem, if the source’s internal impedance is equal to the load impedance, the maximum power is transmitted from source to load. This is accomplished by placing an impedance matching circuit between the segment of the antenna and the rectifier. The impedance matching depends on the specifically designed frequency bands, whenever the source and load is a reactive component. Thus, the RF bandpass filter section is considered as an impedance matching network. In [38], it was demonstrated that the most frequently used RF filters (as an impedance matching network) are L, π and T, respectively. The proposed RF bandpass filters circuit is a modification of a T-network RF filter. A capacitor is placed between two parallel inductors and two inductors at either end (i.e., forming a modified T-network) to increase the bandwidth and restrain unwanted high-frequency bands near the resonance band [39]. Moreover, additional matching elements, i.e., a radial stub is inserted between the RF filter and rectifier section, as shown in Figure 14.

These matching elements’ function is to continue the rectifier’s performance through a broad range of load resistance. Radial stub is a planar element that comprises of a segment of a circle instead of a constant-width line. It is applied with planar transmission lines for a low impedance stub requirement. When the junction between the main line and a stub wide line is not a perfectly-defined point, it is used as the point to maintain the impedance and increase the bandwidth. To cover the desired frequency bands, the chosen BW is 0.22 GHz. A single-band RF bandpass filter (i.e., impedance matching circuit) is designed to adapt to a modified rectifier’s input impedance to a 50 Ω resistive port. The matching network (with the combination of four inductors and a single capacitor) is designed to cover GSM 900 by first RF filter network, GSM 1800 by second RF filter network, 3G and Wi-Fi by third and 4th RF filter network, respectively. Figure 15 shows the complete quad-band rectifier topology. The SMD Murata components (replacement of lumped components) are used for the actual product model to achieve the best accuracy. The EM solver of ADS is used to analyse the dielectric and insertion loss of the substrate and microstrip line, respectively. Moreover, the optimized values of all components (i.e., Table 3) of the four RF bandpass filters are also determined by the EM optimization solver of ADS software. The complete layout and fabricated prototype image of the proposed multiband rectifier are depicted in Figure 16a,b, respectively. The substrate is FR4 with a dielectric constant of 5.4 and a height of 1.6 mm.

## 6. RF Simulation Setup and Performance Analysis

The wideband multi-frequency log-periodic antenna is used as a power generator for various frequency tones.

Four different and simple capacitors (i.e., 680 pF, 2.4 pF, 0.6 pF and 220 pF) are used as low-pass filters for different rectifier branches. To feed the sensor, an equivalent impedance is used for the RF harvester load model. Generated dc voltages of the corresponding load level for recently designed sensors are summarized by Table 4. The lowest possible generated voltage trend is very near to the threshold voltage (i.e., about 0.35 V). Their load level varies from 3 kΩ to 25 kΩ. Two realistic load resistance values, such as 6.81 kΩ and 7.5 kΩ, are selected due to higher performance. The simulated and measured S-parameters of the proposed multiband rectifier at six input RF power levels are shown in Figure 17. It can be observed that the suggested multiband rectifier covers the four bands (i.e., which are available at ambient level) for different RF input power density levels.

These results (i.e., comparison between simulated and measured reflection coefficient) confirm an excellent correlation at the frequency bands around 0.9 GHz, 1.8 GHz, 2.12 GHz and 2.4 GHz for −5 dBm, −10 dBm, −15 dBm, −20 dBm and −27dBm, while the S11-parameter values at 0.9 GHz, 2.4 GHz and 1.84 GHz have shifted to lower and upper frequencies to cover a higher bandwidth for the −35 dBm RF input power level. The excellent impedance matching (i.e., four parallel RF filter networks) is achieved at the different available frequency bands, such as 0.9 GHz, 1.84 GHz, 2.12 GHz and 2.4 GHz, respectively. This behaviour is probably because of utilizing the realistic SMD elements in the multiband rectifier circuit. The value of SMD capacitors and inductors are a function of frequency. Different series of SMD components with large values (i.e., as best as possible) are used instead of similar products with lower rectifier circuit values to decrease the parasitic effect. The comparison between simulated and measured RF-to-DC rectification efficiency at four different frequencies with three different associated power density levels which is a function of load resistance is presented in Figure 18. Three other frequency inputs with corresponding three RF power density levels in this scenario are 0.9 GHz, 1.84 GHz, 2.12 GHz and 2.4 GHz, −20 dBm, −27 dBm and −35 dBm, respectively.

It is shown that the proposed quad-band rectifier works rigorously over a vast range of load resistance (i.e., between 5 kΩ to 10 kΩ) at all available frequency bands with associated RF power density levels indicating that the variation of load effects has been decreased. In Figure 18, it is observed that dc rectification efficiency is larger than 80% (at 0.9 GHz for −20 dBm input power level), 50% (at 1.84 GHz for −20 dBm input power level), 55 % (at 2.12 GHz for −20 dBm input power level) and 40% (at 2.4 GHz for −20 dBm input power level), respectively, for the load value between 6 kΩ and 8 kΩ (more especially). The dc conversion efficiency is higher than 66% (at 0.9 GHz, −35 dBm RF input), 42% (at 1.84 GHz, −35dBm RF input), 52% (at 2.12 GHz, −35 dBm RF input) and 32% (at 2.4 GHz, −35 dBm RF input) for a load resistance ranging from 6.18 kΩ to 7.5 kΩ. The rectifier’s performance is continued for a variety of load values that are suitable in various ambient applications. The dc conversion efficiency versus RF input power level of the proposed rectifier is given in Figure 19. It is observed that the measured RF-to-DC conversion efficiency of the proposed harvester increases as the number of tones is increased. The maximum dc rectification efficiency at a single tone (at 0.9 GHz) is around 35% with −20 dBm RF input and has increased overall efficiency of above 17% for a cumulative of four RF tones. This is due to an excellent impedance matching between the multiband antenna and rectifier for all respective RF branch, and the resultant dc voltage is satisfactory.

Alongside the impedance change, therefore, the RF-to-DC conversion efficiency is not optimum if some of the incident powers are null or do not have the same value. The comparison between simulated and measured DC rectification efficiency versus frequency for different power density levels is shown in Figure 20. It is shown that the RF-to-DC rectification efficiency is more significant at lower frequency bands (0.8 GHz to 1GHz) and smaller at higher frequency bands (1.8 GHz to 2.5 GHz) for the same RF power density level. This occurred due to parasitic elements’ losses in a particular diode and FR4 substrate material for higher frequency regions. The simulation is conducted for the −20 dBm available RF input power and depicts a 6% enhancement of each sub-band’s peak efficiency. Finally, the generated dc voltage of the proposed rectifier for multi-tone RF input signals has been examined. The comparison between simulated and measured output dc voltage as a function of RF input power density level for multi-tones is shown in Figure 21. It is shown that with the four sources turned on, the dc output voltage is almost equal to the dc total of every dc input of the four RF tones. The circuit is connected to four different RF power generators, with a respective transmission frequency of 0.9 GHz, 1.84 GHz, 2.12 GHz and 2.4 GHz. The measured dc output voltage is determined as a function of ambient RF input power source for the first, second, third and fourth frequency signal. In this sense, the circuit generated resultant dc voltage has been verified in the presence of simultaneous multiple tones.

## 7. Real Phase Measurements in RF Harvester

The measurement setup and the instruments such as Anritsu MS2024A VNA Master handheld vector network analyzer and Tektronix TSG4104A Series RF vector signal generator are used to verify the performance of the proposed RF harvester prototype. The measurement of the prototype is conducted in two different conditions.

### 7.1. Employing Known Frequency Signals and RF Power Source

A multi-frequency broadband log-periodic antenna is connected to an Anritsu MS2024A Master handheld vector network analyzer (VNA) for verifying the performance compared to the simulated result. The TSG4104A RF vector signal generator is connected with a multiband rectifier to realize the generated dc voltage by transmitting available frequency signals within the frequency range of 0.5 to 3 GHz signals with a 10 MHz step size.

The TSG4104A signal generator is used to input specific frequency signals (available in ambient environment) with the associated RF power level (ambient RF power density level, i.e., −35 dBm to −10 dBm) environment) and a digital multimeter is connected to record the generated dc output voltage. Figure 22 shows the generated 0.354 V (Figure 22a) dc output voltage for 0.97 GHz frequency signal with −35 dBm RF power density level. With the same frequency signal and −27 dBm associated RF power level produced, the dc output voltage reached 0.402 V (Figure 22b). Similarly, generated dc voltage is recorded for different frequency signals (i.e., 1.84 GHz, 2.12 GHz and 2.4 GHz) with RF power density levels of −35 dBm and −27 dBm (since that is the available RF power level in an ambient environment). It is demonstrated that the recorded dc voltage for each frequency with the associated RF power level is reasonable to power any micro-electronic devices.

### 7.2. Using a Random and Unpredictable (i.e., Ambient) Energy Source

To assess the proposed RF harvester’s realistic performance, a typical ambient outdoor environment with a very low RF power density is selected to measure the performance. In Figure 23a, the RF harvester is placed in an outdoor environment, and a digital multimeter is directly connected to the rectifier load branch for measuring the dc output voltage. The recorded dc output voltage varies from 0.350 V to 0.687 V. The harvested dc output voltage enhanced 0.720 V with the integration of evaluation modules (EVMs) which is shown in Figure 23b. The dc output in dBm can be determined by
(3)$$P_{dc}(dBm)=10\log_{10}⟮\frac{V_{dc}{}^{2}}{R_{load}}\times 10^{3}⟯$$

Here, $P_{dc}$, $V_{dc}$ and $R_{load}$ are received power in dBm, output voltage (V) and load resistance, respectively. The measured dc output voltage is converted into dBm by using Equation (3), and it is found to be ranging from −25 dBm to −23 dBm, which is larger than the available ambient power density level of −35 dBm to −27dBm. This is due to the broadband harvester with united received RF power over all frequency bands into dc power, the resultant RF power is, therefore, greater than the input RF at a specific band. The overall power rectification efficiency can also be determined by dividing the average resultant received RF power in each band by the dc output voltage. The maximum efficiency can be achieved with the load resistance of 6 kΩ to 7.81 kΩ.

Table 5 shows the comparison between the proposed harvester and several related works. It is observed that most of the related work is for single, dual, triple and multiband RF harvesters. The overall power rectification efficiency can also be determined by dividing the average resultant received power in each band by dc output voltage. The maximum efficiency can be achieved with the load resistance between 6 kΩ to 7.81 kΩ. Table 5 shows the comparison between the proposed harvester and several related works. It is observed that most of the related work is for single, dual, triple and multiband RF harvesters. The proposed work is a quad-band RF harvester with a higher dc rectification efficiency and generated dc output voltage. The dimension of the new rectenna is relatively smaller than the majority of previous harvesters. Moreover, the majority of previous designs are achieved higher conversion efficiency with higher RF power (i.e., 0 dBm,−10 dBm, −15 dBm and so on) as well as output dc generated voltage but the new design is suitable for ambient RF power level (i.e., −20 dBm, −27dBm, −35dBm) in terms of efficiency and dc output voltage.

## 8. Conclusions

A highly efficient quad-band RF harvester is proposed for ambient RF energy harvesting. A novel impedance matching network is introduced with novel broadband rectifying components to better align the usable RF signals with comparatively lower RF power density levels. The highly sensitive full-wave rectifier circuit is intended to strengthen the low power sensitivity. A broadband multi-frequency self-complementary log-parodic antenna is also installed in an ambient setting to improve the RF signal’s receiving power. The harmonic rejection potential is embedded with the proposed antenna by using front-to-front arms with multiple teeth structures to optimize the overall rectification efficiency and retain the overall dimension as compact as possible. The simulated and measured performances have shown that the harvester has a maximum dc rectification efficiency of about 52 percent for −20 dBm input RF power from 0.9 GHz to 2.6 GHz. The rectified output dc voltage can be well beyond the incident RF power from any single sound because of the wideband activity and highly sensitive nature. Given the above performances (i.e., dc rectification efficiency, output dc voltage) of this rectenna in a comparatively low RF power density ambient environment, this harvester can be used for ambient wireless RF energy scavenging for a huge number of wireless devices and network applications.

## Figures and Tables

**Figure 1 sensors-22-00424-f001:**
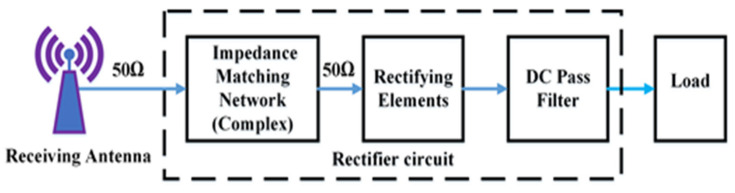
Basic block diagram of RF energy harvesting technology.

**Figure 2 sensors-22-00424-f002:**
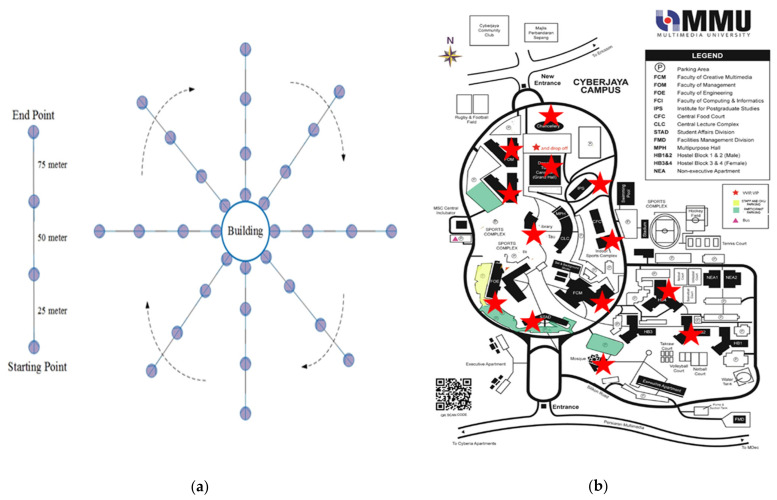
(**a**) The proposed strategy of RF spectral survey for each survey point, (**b**) map of the MMU area indicated RF survey points [21].

**Figure 3 sensors-22-00424-f003:**
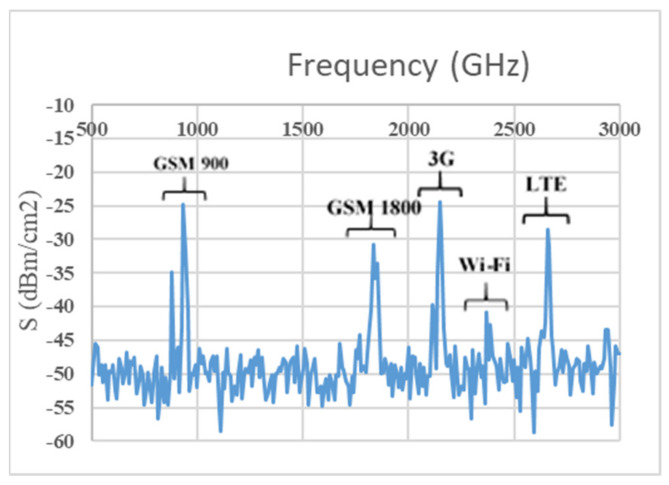
Measurements of available frequency bands and associated RF power levels.

**Figure 4 sensors-22-00424-f004:**
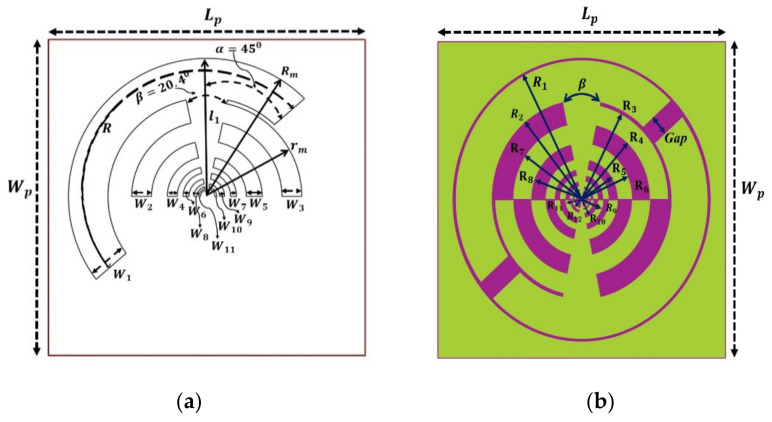
Geometry of the LP antenna, (**a**) detailed view of the single-sided structure, (**b**) complete design with dimensional details.

**Figure 5 sensors-22-00424-f005:**
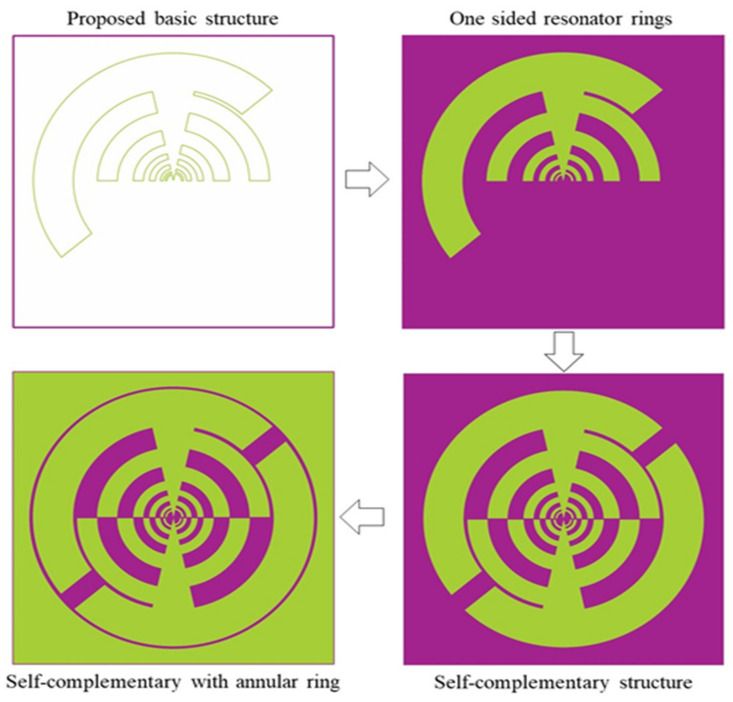
The proposed antenna’s sequential design process, a two-armed log-periodic self-complementary dual-arm teeth structure with a separate port and annular ring shape.

**Figure 6 sensors-22-00424-f006:**
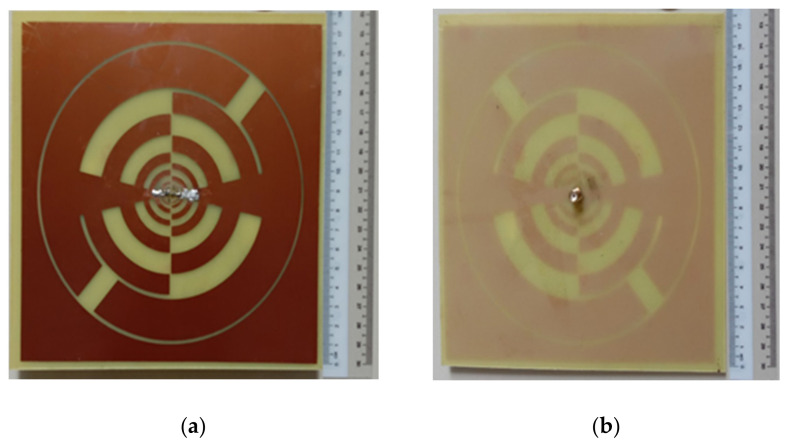
Fabricated prototype (**a**) front side (**b**) backside of the proposed antenna.

**Figure 7 sensors-22-00424-f007:**
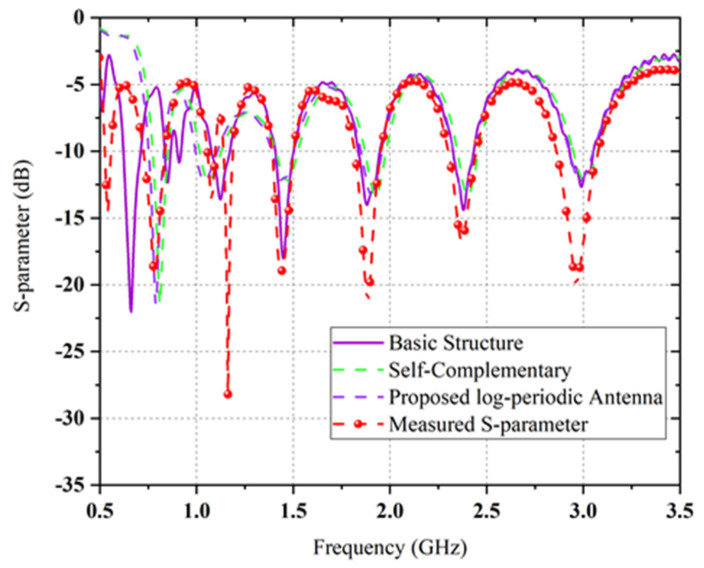
The comparison between simulated and measured scattering parameter (S11) of the proposed antenna.

**Figure 8 sensors-22-00424-f008:**
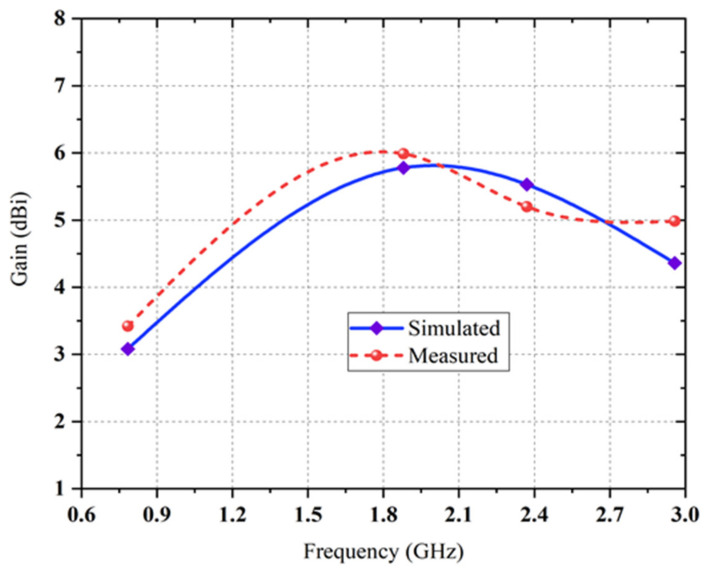
The comparison between simulated and measured peak realized gain of the proposed antenna.

**Figure 9 sensors-22-00424-f009:**
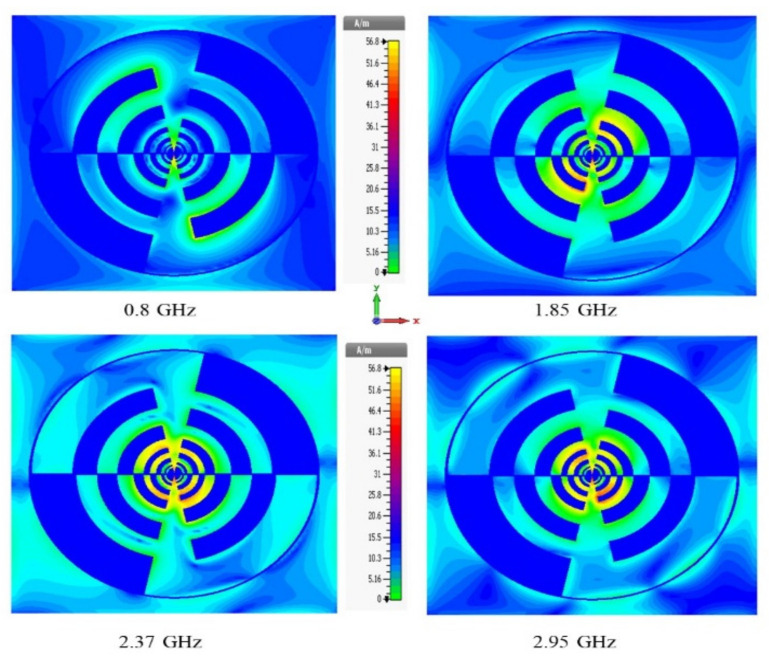
The surface current distribution at operating frequency bands (i.e., 0.78 GHz, 1.8 GHz, 2.38 GHz and 2.9 GHz) on the novel rings log-periodic antenna.

**Figure 10 sensors-22-00424-f010:**
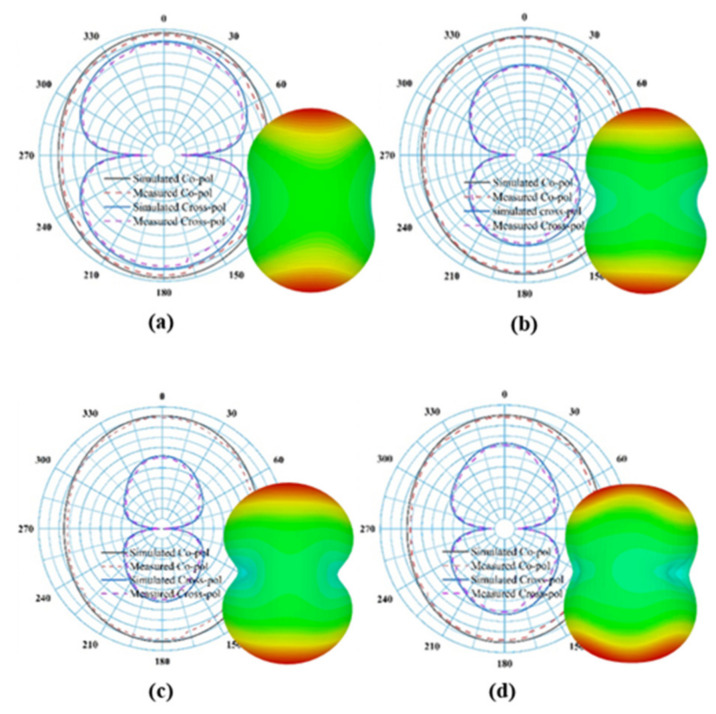
A 2D (measured and simulated) and a 3D (measured) radiation patterns for the dual arm-LP antenna for different frequencies such as (**a**) 0.78 GHz, (**b**) 1.8 GHz, (**c**) 2.38 GHz and (**d**) 2.9 GHz.

**Figure 11 sensors-22-00424-f011:**
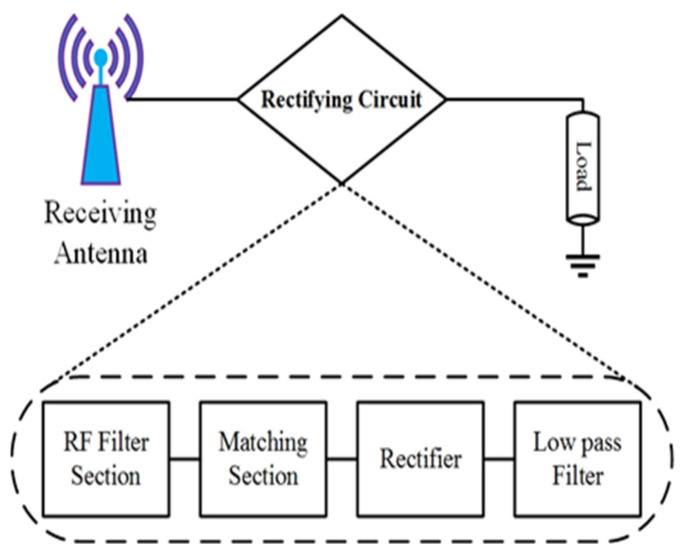
Multi-frequency RF harvesters design with elementary details in rectifier architecture.

**Figure 12 sensors-22-00424-f012:**
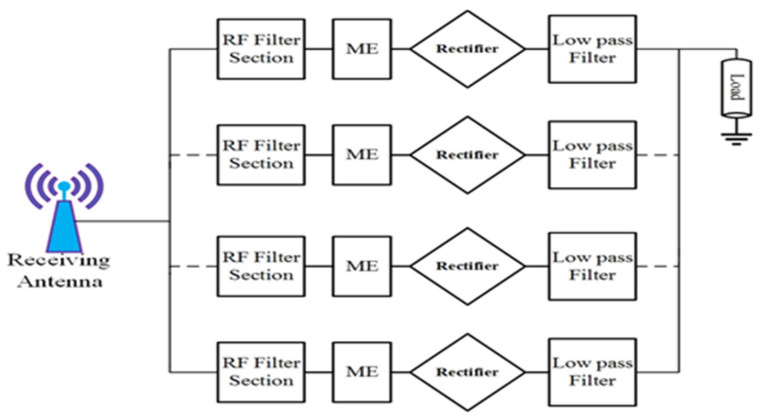
Proposed multiband RF harvesters topology (ME = matching element).

**Figure 13 sensors-22-00424-f013:**
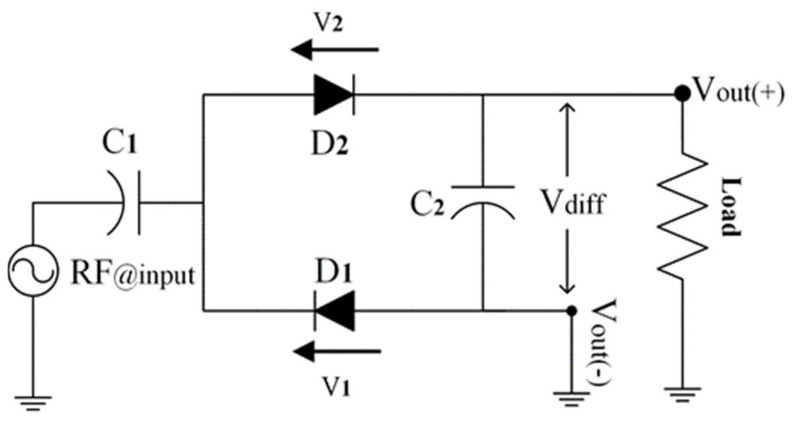
Proposed rectifier circuit.

**Figure 14 sensors-22-00424-f014:**
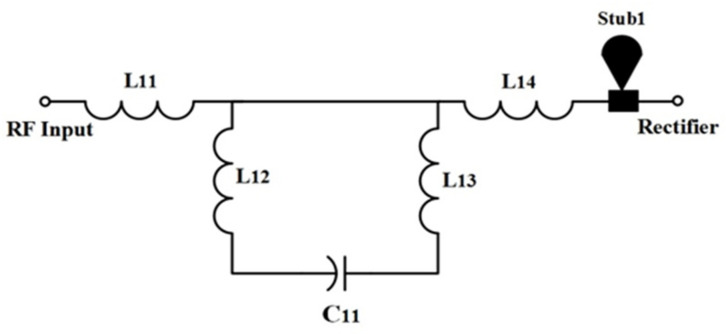
The single RF filter (matching network) including a microstrip matching element for maintaining loads ranging from 0.5 kΩ to 25 kΩ for the RF input power levels of −35 to −10 dBm.

**Figure 15 sensors-22-00424-f015:**
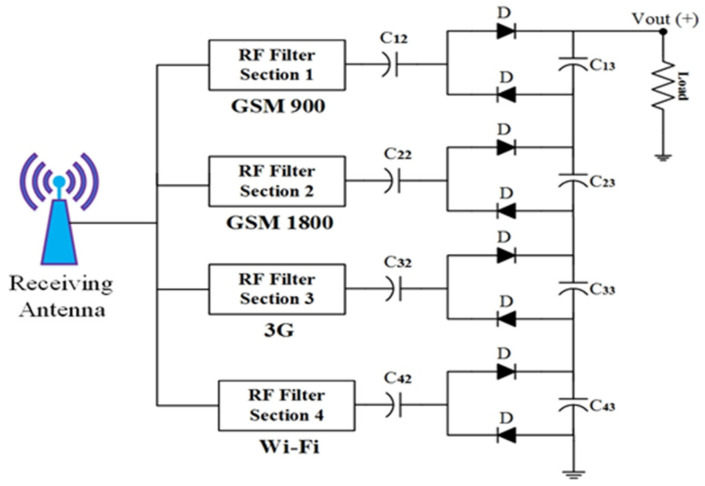
The complete rectifier topology.

**Figure 16 sensors-22-00424-f016:**
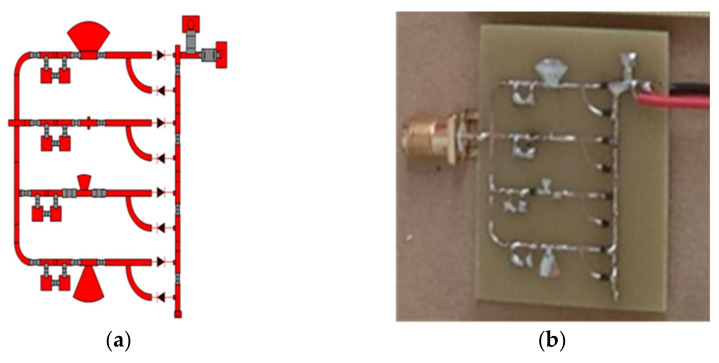
The optimized (**a**) layout and (**b**) prototype image of the four-band rectifier topology. The active area of the PCB is 32.07 mm × 35.75 mm.

**Figure 17 sensors-22-00424-f017:**
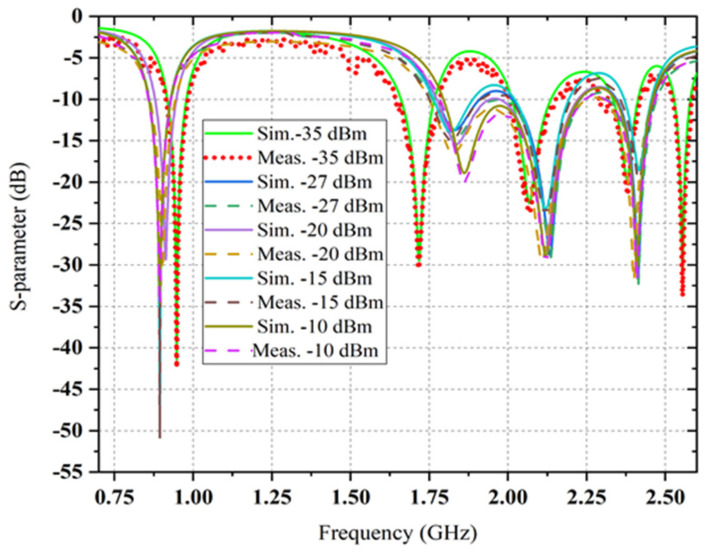
The comparison between simulated and measured scattering parameters of the suggested multiband rectifiers at six available ambient input RF power density levels for a constant load of 6.81 kΩ.

**Figure 18 sensors-22-00424-f018:**
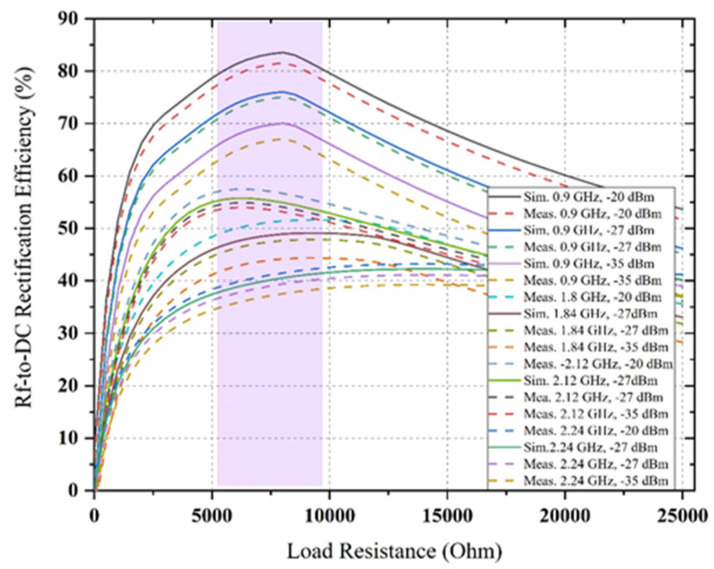
The comparison (between simulated and measured) of RF-to-DC rectification efficiency of the rectifier circuit as a function of load resistance for different frequencies with associated RF power levels.

**Figure 19 sensors-22-00424-f019:**
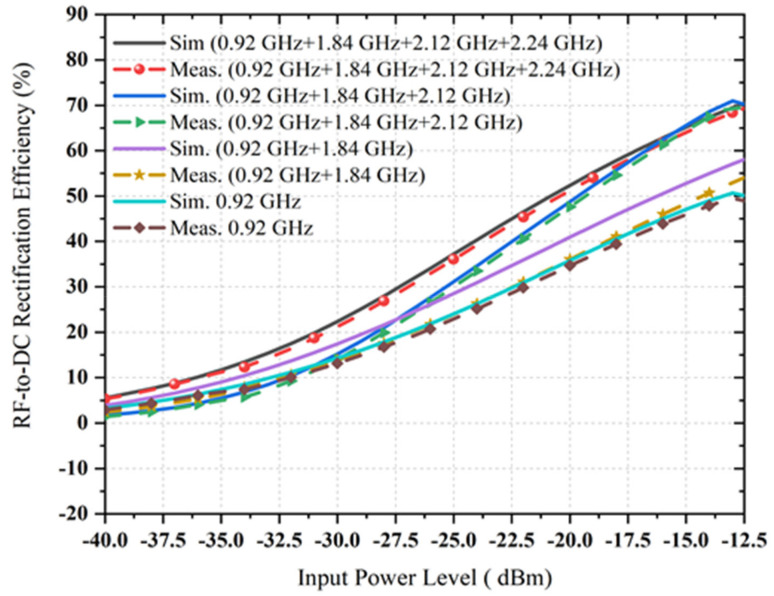
The comparison between simulated and measured RF-to-DC rectification efficiency versus RF input power level of the new rectifier at one, two, three and four frequency tones for the load resistance of 6.81 kΩ.

**Figure 20 sensors-22-00424-f020:**
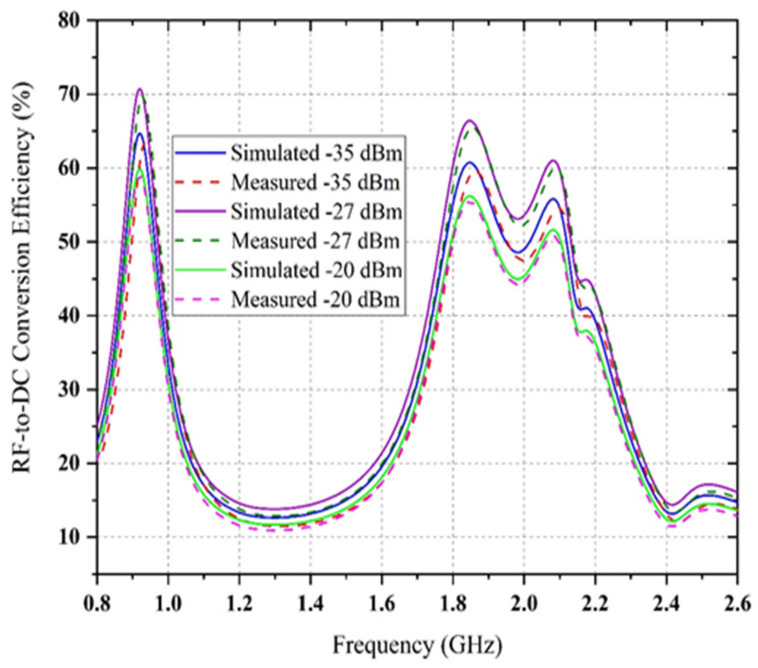
The comparison between simulated and measured DC rectification efficiency of the quad-band rectifier versus frequency at four RF input power levels and a constant load of 6.81 kΩ.

**Figure 21 sensors-22-00424-f021:**
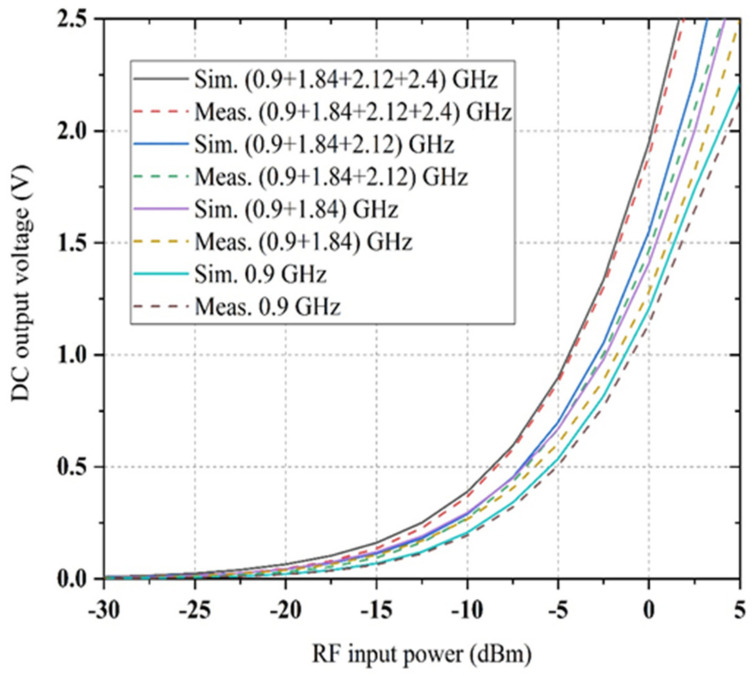
The comparison between simulated and measured output voltage versus input RF power for several tones.

**Figure 22 sensors-22-00424-f022:**
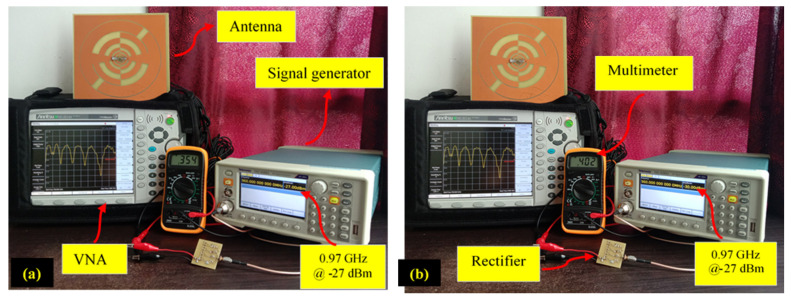
The measurement setup for antenna and rectifier in lab environment (**a**) for −35 dBm RF input (**b**) for −27dBm RF input.

**Figure 23 sensors-22-00424-f023:**
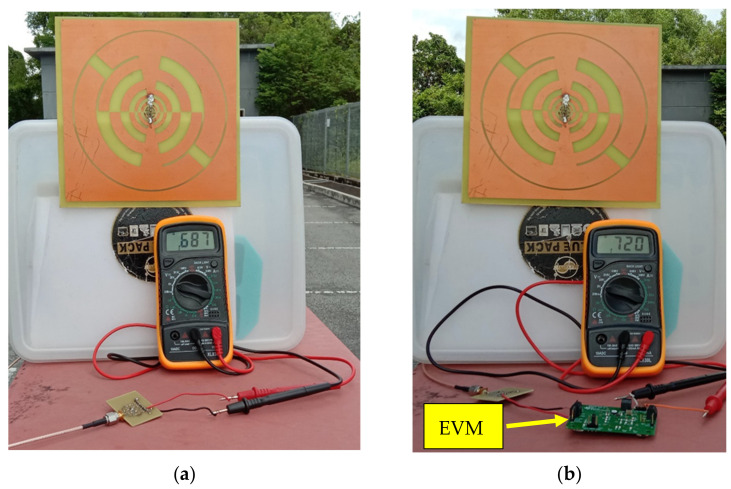
Measurement in the ambient environment (**a**) harvester only (**b**) harvester with EVM.

**Table 1 sensors-22-00424-t001:** Summary of RF survey at MMU.

Brand Name	Frq. Range (GHz)	Max. Power (dBm)
GSM 900 (UpLink)	0.88–0.915	−27.6
GSM 900 (DownLink)	0.925–0.960	−22.5
GSM 1800 (UpLink)	1.71–1.785	−31.5
GSM 1800 (DownLink)	1.805–1.880	−17.8
3G (UplinkL)	1.920–1.980	−27
3G (DownLink)	2.110–2.230	−23
Wi-Fi	2.39–2.40	−25.6
LTE	2.50–2.70	−26.5

**Table 2 sensors-22-00424-t002:** Log-periodic antenna design parameters.

Parameter	Value (mm)	Parameter	Value (mm)
$L_{p}$	160	$R_{1}$	69.72
$W_{p}$	160	$R_{2}$	49.82
$W_{1}$	20	$R_{3}$	47.79
$W_{2}$	10	$R_{4}$	37.85
$W_{3}$	10	$R_{5}$	11.87
$W_{4}$	5	$R_{6}$	19.86
$W_{5}$	8	$R_{7}$	37.92
$W_{6}$	3	$R_{8}$	27.89
$W_{7}$	3	$R_{9}$	19.93
$W_{8}$	1.5	$R_{10}$	14.91
$W_{9}$	2.5	$R_{11}$	3.84
$W_{10}$	1	$R_{12}$	3.05 m
$W_{11}$	0.8	Gap	0.5
l	69.76	α	45°
R	70	β	20.4°

**Table 3 sensors-22-00424-t003:** Optimized value of all components for proposed rectifier topology.

Parameter Symbol	Filter Section 1 *i* = 1	Filter Section 2 *i* = 2	Filter Section 3 *i* = 3	Filter Section 4 *i* = 4
Li1	22 nH	1.5 nH	3.3 nH	3.9 nH
Li2	27 nH	5.5 nH	1.2 nH	150 nH
Li3	27 nH	8.2 nH	1.5 nH	1 nH
Li4	1 nH	2.4 nH	6.2 nH	3 nH
Ci1	1.8 pF	0.2 pF	3.6 pF	0.3 pF
Ci2	33 pF	8.2 pF	220 pF	8.2 pF
Ci3	680 pF	2.4 pF	0.6 pF	220 pF
Wi1	2.5625 mm	0.3125 mm	0.9375 mm	0.875 mm
Lstubi1	2.47 mm	0.3 mm	1.4775 mm	3.565 mm
βi1	84.7°	20°	38°	56.6°

**Table 4 sensors-22-00424-t004:** Typical harvesters generated dc voltage for the corresponding load.

Ref. No	[40]	[41]	[42]	[43]	[44]	[45]	[46]	[47]	[48]
Load (kΩ)	3	7	9	9.53	11	12	12	14	24.3
DC (V)	0.41	1	0.32	0.65	0.8	0.49	0.24	0.45	0.3

**Table 5 sensors-22-00424-t005:** Comparison between the proposed harvester and some related works.

Ref. No.	Number of Bands	Frequency Bands (GHz)	Dimension (mm)	RF Input Power Level (dBm)	Maximum DC Rectification Efficiency (%) @ dBm	Max. Conversion Efficiency (%)	DC Voltage (V)	Load (kΩ)
[6]	Dual-band	1.8, 2.2	300 × 380 × 1.6	−5 to −30	55 @ −30	55	N/A	5
[5]	Dual-band	0.915, 2.4	61.5 × 48 × 0.025	−11.6 to −13.7	56.2 @ −11	56.2	N/A	2.2
[13]	Dual-band	1.8, 2.1	70 × 70 × 13.2	−10 to −30	55 @ −10	70	0.298	15
[16]	Quad-band	0.9, 1.75, 2.15, 2.45	155 × 155 × 7.2	−15 to 0	60 @ 0	60	2.500	N/A
[12]	Quad-band	0.9, 1.8, 2.1, 2.4	100 × 100 × 1.6	−25 to 0	65@ 0	84	0.409	11
[30]	Quad-band	0.55, 0.9, 1.85, 2.15	N/A	−10 to −29	40 @ −12	40	N/A	N/A
[1]	Hexa-band	0.55, 0.75, 0.9, 1.85, 2.15, 2.45	160 × 160 × 1.6	−5 to −30	67 @ −5	80	0.663	10~75
**New**	**Quad band**	**0.9, 1.8, 2.12, 2.4**	**160 × 160 × 1.6**	**−10 to −35**	**52 @ −20**	**52**	**0.687**	**6.18~7.5**

## Data Availability

Not applicable.

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
