# Peer review of "Design of a Highly Efficient Wideband Multi-Frequency Ambient RF Energy Harvester"

_sensors, 2022, doi:10.3390/s22020424_

Round 1
Reviewer 1 Report
In this article, a highly efficient quand-band RF harvester (over the frequency range from 0.8GHz to 2.6GHz) is fully presented for ambient, low input RF power density levels utillization (as low as -20dBm) by an improved design of impedance matching network (IMN) with T-network RF filters. The proposed system implementation constitutes a realistic solution for conversion efficiency incrementation at limited RF power range of ambient environment.
The self-complementary log-periodic broadband antenna design on planar FR4 substrate is well presented for use in quand band (0.78GHz, 1,8GHz, 2,38GHz, 2,9GHz). The measurement results are clearly presented and depicts that the harvester is able to harvest dc voltage up to 0.67V with 52 percent conversion efficiency (PCE) in ambient RF environment (both indoor and outdoor) at MMU University in Malaysia.
Please check the FR4 relative permittivity substrate value (er) at lines 160 & 413. Note that standard er value for FR4 substrate is 4,35.
Author Response
Thank you for your valuable comments. I have updated the papers according your advice. Pls check attachment.

Reviewer 2 Report
The manuscript presents a low input radio frequency power energy harvester with an improved impedance matching network which overcome the poor conversion efficiency and limited RF power range of the ambient environment of the compared radio frequency power energy harvesters for semi-urban region where are dominant the frequency bands in the 0.8 GHz to 2.6 GHz frequency bands. Firstly, the rectifying circuit's with high efficiency in unpredictable conditions is proposed. Secondly, a
self-complementary log-periodic higher bandwidth antenna is proposed. Then, the design and the test of the proposed RF harvester’s prototype are carried out and the results show the 52 percent efficiency at -20 dBm signal and up to 0.678 V output at approx. 6 KΩ load.
The manuscript is well written and presents interesting results, but I would like to advice authors to make some rearrangements of the text and improve the readability as described bellow:
- The editors in the journal Sensors strongly encourage authors to use the following style of structured abstracts, but without headings: (1)Introduction: Place the question addressed in a broad context and highlight the purpose of the study; (2) Methods: briefly describe the main methods or treatments applied; (3) Results: summarize the article's main findings; (4) Discussion and/or conclusions: indicate the main conclusions or interpretations. So, I would like to advice authors to rewrite the paper to match above mentioned rules.
- Page 12, line 392, Figure 14(a): I think that figure 14(a) is not necessary to be displayed, it is enough to be mention within the text in lines 388 and 389.
- Page 13, line 397-402: Would you describe more precisely the function of stubs and how their dimensions were calculated?
- Page 14, line 421, Table 3: Would you describe in the text what is the parameter βi1?
- Page 19, line 554, Table 5: Make the description of the new harvester bold completely, so it would be more visible and more easier to compare with referenced harvesters.
Author Response

(The authors gave the same response as above.)
